# Clinic and Home-Based Exercise with Blood Flow Restriction Resolves Thigh Muscle Atrophy after Anterior Cruciate Ligament Reconstruction with the Bone-Patellar Tendon-Bone Autograft: A Case Report

**DOI:** 10.3390/healthcare11131885

**Published:** 2023-06-29

**Authors:** Braidy S. Solie, Garrett G. Eggleston, Nicole A. Schwery, Christopher P. Doney, Michael T. Kiely, Christopher M. Larson

**Affiliations:** 1Training HAUS, 2645 Viking Circle, Suite #200, Eagan, MN 55121, USA; 2Twin Cities Orthopedics, 4010 West 65th Street, Edina, MN 55435, USA; chrislarson@tcomn.com

**Keywords:** ACL, blood flow restriction, bone–patellar tendon–bone autograft, practical blood flow restriction, rehabilitation

## Abstract

Anterior cruciate ligament reconstruction (ACLR) results in thigh muscle atrophy. Of the various interventions proposed to mitigate thigh muscle atrophy, exercise with blood flow restriction (BFR) appears safe and effective. Some literature suggests daily exposure to exercise with BFR may be indicated during the early phase of ACLR rehabilitation; this case report outlines the methodology utilized to prescribe clinic- and home-based BFR within an outpatient rehabilitation program. A 15-year-old male soccer player suffered a left knee injury involving the anterior cruciate ligament and both menisci. He underwent ACLR and completed exercise with BFR as part of his clinic- and home-based rehabilitation program, which included practical blood flow restriction during home-based rehabilitation. After 16 weeks of rehabilitation, surgical limb thigh girth values were objectively larger than the non-surgical limb (surgical, 52.25 cm; non-surgical 50 cm), as well as the multi-frequency bioelectrical impedance analysis of his lower-extremity lean body mass (surgical limb, 10.37 kg; non-surgical limb, 10.02 kg). The findings of this case report suggest that the inclusion of clinic- and home-based BFR within an outpatient rehabilitation program may be indicated to resolve thigh muscle atrophy early after ACLR.

## 1. Introduction

Complete disruption of the anterior cruciate ligament (ACL) of the knee results in increased anterior tibial translation at the tibiofemoral joint [1,2]. To reduce the risk of recurrent knee instability and trauma [3,4], anatomic anterior cruciate ligament reconstruction (ACLR) is the preferred surgical treatment to facilitate a safe return to high-demand physical activity [5,6,7]. Regarding surgical technique, ACLR with a bone–patellar tendon–bone (BPTB) autograft may yield better graft survivorship in young athletes than a hamstring autograft [8,9,10], with other literature suggesting anatomic repair of the knee’s menisci (if damaged at the time of knee injury) may further optimize knee stability and graft survivorship [11,12,13,14].

After ACLR, significant thigh muscle atrophy (TMA) can occur within as little as 5–16 days [15,16], and may be present well beyond the typical postoperative rehabilitation period (9+ months after ACLR) [17,18]. The reason for TMA after ACLR is multifactorial; joint pain [19], knee effusion [19], neurologic muscle inhibition [19,20], knee range-of-motion deficits [21], and immobilization/disuse of the surgical limb are believed to be causal drivers of the catabolic cascade observed within the thigh muscles [15,22,23,24,25,26]. This cascade results in rapid protein breakdown and affects the anti-gravity muscles of the thigh the most (i.e., the quadriceps muscle group is more affected than the hamstrings) [27,28,29,30].

Considering the high prevalence of TMA after ACL injury [26], various rehabilitation strategies have been proposed to mitigate the catabolic effect of ACLR on the thigh muscles [31,32,33,34,35,36,37], especially the quadriceps [34,35]. Of these strategies, exercise with blood flow restriction (BFR) has become increasingly popular within the rehabilitation setting [35,38,39,40,41,42]. Multiple studies have reported the beneficial effect of BFR on muscle size and strength after ACLR [35,40,43,44]. However, Caetano et al. [45] reported significant heterogeneity in the postoperative exercise prescription parameters for BFR, limiting the ability to draw strong conclusions as to the best way to implement BFR within the various stages of rehabilitation after ACLR.

To stimulate improvements in muscle size and strength, the ideal BFR exercise frequency is reported to be 2–3×/week with an exercise intensity of 20–40% of an individual’s one-repetition max (1-RM) [46]. However, during the first 6–12 weeks of rehabilitation after ACLR, a frequency of 1–2×/day may be more efficacious, as exercise intensity may be limited to a load-intensity <20% 1-RM (due to joint irritability and/or the postoperative healing response) [28,46,47]. Therefore, patients completing outpatient rehabilitation at a frequency of 1–3×/week may not receive adequate exposure to exercise with BFR (within the clinic-setting alone) during the first 2–3 months after ACLR [48,49].

Recently, more practical methods of blood flow restriction have become popular [50]. Practical blood flow restriction (pBFR) uses less expensive equipment, making access to exercise with pBFR more feasible within a non-clinical setting [50]. Moreover, pBFR has shown similar efficacy to more sophisticated BFR devices [51]. Considering exercise frequency and volume are key factors for the maximal stimulation of muscle hypertrophy [45,52,53], the purpose of this case report is to outline the pragmatic BFR + pBFR exercise prescription used to resolve TMA in a young male athlete after ACLR.

## 2. Materials and Methods

### 2.1. Case Presentation

In October of 2021, a 15-year-old male soccer player presented to the treating orthopedic surgeon’s office (C.M.L). The athlete reported suffering a soccer-related left knee injury (non-dominant limb) 6 days prior, which was the result of direct contact to the lateral trunk during competitive play (contact mechanism of knee injury). He reported hearing a “pop” in his knee at the time of injury but was able to immediately bear weight within the involved limb.

### 2.2. Differential Diagnosis

The athlete presented with a non-antalgic gait pattern and was lacking 4 degrees of passive knee extension in supine. Testing of the vascular/sensory/motor status of the injured limb was completed and confirmed to be without impairment [54]. The athlete’s knee had a 2+ sweep test for knee effusion and a grade II-B Lachman test [55,56]. His knee was stable with valgus and varus stress testing at both 0 and 30 degrees of knee flexion [57]. During the McMurray test [58], the athlete experienced pain both medially and laterally within the knee. The athlete had a negative posterior drawer and dial test, and with the collective exam findings, the surgeon suspected an isolated ACL + meniscus injury pattern within the left knee [59,60].

With a contact/perturbation-based mechanism of knee injury [61], normal testing of the posterolateral corner of the knee [60], and stable neurovascular status [54], magnetic resonance imaging (MRI) of the left knee was ordered to evaluate the ACL and menisci. The athlete was diagnosed with an MRI-confirmed complete tear of the ACL, with vertical tears reported within the posterior horn of the medial and lateral menisci. The knee injury was classified as a Sherman type-II ACL tear [62], and both meniscal tears were considered a stable tear pattern involving the red-red zone of the meniscus [63,64].

### 2.3. Treatment Methods

#### 2.3.1. Preoperative Rehabilitation

Considering the athlete played year-round soccer, his family elected to pursue anatomic ACLR as treatment. He completed 4 outpatient rehabilitation sessions before surgery. Preoperative rehabilitation sessions were devoted to athlete education/home programming, the resolution of his knee effusion, improving knee-specific range of motion (ROM), and increasing quadriceps activation/function. Preoperative rehabilitation goals were met before surgery and defined as: (1) trace/1+ knee effusion on the sweep test [55], (2) knee extension/hyperextension ROM equivalent to the contralateral limb [65,66], (3) knee flexion ROM greater than or equal to 120 degrees [66], (4) no lag during the straight leg raise exercise with the knee in terminal knee extension/hyperextension [66,67].

#### 2.3.2. Surgical Technique and Postoperative Precautions

The athlete underwent ACLR 22 days after his knee injury. His ACL was reconstructed with a 10 mm bone–patellar tendon–bone autograft [8,9,10]. His meniscal tears were anatomically repaired utilizing the gold-standard, all-inside technique [63]; the medial meniscus required 3 Fast-Fix Flex™ sutures (Smith & Nephew, London, UK) for anatomic repair, with the lateral meniscus requiring only a single suture.

Rehabilitation began the day after surgery, and all postoperative precautions were implemented in response to the stable meniscal repairs at the time of ACLR. Range-of-motion exercises were prescribed, involving 0–90 degrees of knee flexion for 2 weeks with subsequent progression toward full motion. Weightbearing progressed as tolerated with the knee in extension for 6 weeks. Bilateral axillary crutches and a hinge knee brace were utilized for 4 weeks after surgery to protect the meniscal repairs, with a gradual wean from crutches over the following 2 weeks. Once the athlete was 6 weeks out from surgery, closed-kinetic-chain (CKC) squatting and lunging progressions were protected for an additional 6 weeks between 0–90 degrees of knee flexion (i.e., no squatting/lunging > 90 degrees of knee flexion for 3 months).

#### 2.3.3. Immediate Postoperative Rehabilitation

Clinic-based rehabilitation sessions were completed 2×/week for the first 16 weeks after ACLR (Figure 1). Rehabilitation sessions were 45 min in duration, with an additional 15 min of electrostimulation therapy to the quadriceps muscle at the end of each session [68]. In weeks 0–2, the athlete completed a home rehabilitation program 5–6×/day consisting of both open- and closed-kinetic-chain (CKC) exercises (Table 1) [69]. Open-kinetic-chain (OKC) quadriceps exercises were restricted to the weight of the lower leg within 0–90 degrees of knee flexion for 4 weeks to facilitate graft osteointegration within the bone tunnels [70]; quadriceps setting in terminal knee extension was advanced towards the straight leg raise and sitting knee extension exercises. Sustained isometric contractions for 45 s in duration were prescribed during quadriceps training to improve neuromuscular activation within the surgical limb (Table 1) [68,71]. The “crutch quad set” exercise was used as an early phased CKC exercise regression and was advanced by progressively increasing the level of elastic band resistance as joint homeostasis and pain allowed (Figure 2). The sweep test and authors’ version of a modified pain-monitoring scale were used throughout rehabilitation to guide the advancement of all exercise prescriptions (Figure 1) [55,68,72]. Additional quadriceps activation exercises were completed 2×/day with a metronome and home electrostimulation garment (Neurotech^®^, Kneehab XP™) (Table 1) [73,74].

#### 2.3.4. Clinic-Based Exercise with Blood Flow Restriction

Starting in postoperative weeks 1–2, exercise with BFR was introduced during clinic-based rehabilitation sessions (Table 1). The athlete was risk stratified prior to implementing exercise with BFR and found to be at a low-to-moderate risk of an adverse event (based solely on his recent history of arthroscopic surgery) [75]; athlete and parental consent was received prior to implementing BFR within the rehabilitation plan of care. Exercise with BFR was prescribed utilizing the Delphi Personalized Tourniquet System (Delphi Medical, Vancouver, BC, Canada). A 10 cm-wide, Easy-Fit Tourniquet (Delphi Medical) was placed around the most proximal aspect of the athlete’s thigh [76], and an arbitrary tourniquet pressure of 180 mmHg was used to elicit partial vascular occlusion during the athlete’s initial exposures to BFR [28,47]. The Rate of Perceived Tightness (RPT) scale was used to progress torniquet pressure from 180 mmHg to a 7–8/10 RPT over the course of 2 rehabilitation sessions (Figure 2) [77,78,79,80]; the athlete reported a 7–8/10 RPT at 210 mmHg, which was the torniquet pressure applied throughout the remainder of his clinic-based rehabilitation period. To assess blood flow distal to the tourniquet, a vascular doppler was applied to the tibialis posterior artery to confirm that arterial blood flow was maintained at rest (Figure 3a) [79], and a great toe capillary refill time < 3 s was used as an indirect measure of incomplete vascular occlusion (i.e., <100% arterial occlusion with the tourniquet inflated to exercising pressure) (Figure 3c–e) [81,82].

Exercise with BFR was first implemented with light, elastic band resistance or active range of motion exercise with the goal of building to 20–30 min of total occlusion time per exposure (i.e., 4–6 sets collecting 3–5 min of occlusion per set) (Table 1) [28,83]. From weeks 5–7 after ACLR, bodyweight and progressive resistance exercise was permitted/tolerated; sets to volitional fatigue were then utilized to reflect best evidence for hypertrophy-based resistance training [84]. The set volume for the quadriceps muscle(s) progressed from 8 sets/week (postoperative weeks 2–4) to 12+ sets/week (postoperative weeks 5+) of OKC+CKC exercise [84].

#### 2.3.5. Home-Based Exercise with Practical Blood Flow Restriction

Starting in postoperative weeks 3–4, pBFR was introduced within the athlete’s home-based rehabilitation program. Exercise with pBFR was implemented with a 5 cm-wide, non-elastic tourniquet (RockCuff Inc, South Jordan, UT, USA), which was purchased by the athlete’s family (Figure 2). The RockCuff tourniquet was placed around the most proximal aspect of the athlete’s thigh and secured by the device’s ratchet-based cable tensioning system (MOZ reel-knob lacing system). The RPT scale was used as previously published to prescribe partial vascular occlusion during exercise with pBFR [77,78]; the athlete was instructed to tighten the tourniquet to a 7–8/10 RPT prior to the prescribed exercise and completely release the tourniquet in between sets (Figure 2). Using the same methodology as previously mentioned, a vascular doppler was used during an initial, in-clinic instructional session to ensure partial vascular occlusion with the tourniquet tightened to a 7–8/10 RPT (Figure 3a) [79]. The athlete was also advised to ensure a great toe capillary refill time of <3 s when performing pBFR (Figure 3c–e) [81,82].

The pBFR exercise prescription was dosed similarly to a prior high-frequency BFR publication (2 exposures/day) with the goal of mitigating TMA and stimulating an anabolic/anti-catabolic physiological response (i.e., cellular swelling) within the surgical limb [28,83]. The athlete was instructed to perform sets to volitional fatigue [46,84], with the goal of collecting 20–30 min of occlusion time each exposure [28]. Exercise selection for pBFR followed a similar progression as the clinic-based BFR exercise prescription, with the progression of resistance-band and active range-of-motion exercise on to body-weight exercise (Table 1). However, exercise selection was less progressive due to the equipment constraints of home-based rehabilitation, and active range-of-motion exercise was a staple feature of the pBFR exercise prescription. Set volume was standardized to 6 sets/exposure and included OKC+CKC exercise for the thigh muscles of the surgical limb. Home program compliance was monitored/tracked by verbal confirmation and the testing of home-based pBFR setup and exercise techniques within the weekly, clinic-based rehabilitation sessions.

#### 2.3.6. Yielding/Holding Isometric Exercise Program

Starting in postoperative weeks 6–7, the athlete’s exercise prescription was adjusted to include a yielding (“holding”) isometric exercise program targeted at the patellar tendon of the surgical knee (Figure 1) [85]; isometric resistance exercise has been observed to be a beneficial exercise mode for the acute and chronic management of patellar tendinopathy [86,87]; and with the use of the bone–patellar tendon–bone autograft for this athlete’s ACLR, isometrics were implemented during clinic and home-based rehabilitation to improve the load-capacity of the patellar tendon.

The base regression for the yielding isometric exercise prescription was the double leg squat exercise; the athlete was instructed to perform the squat with a narrow base of support to mitigate inter-limb off-loading onto the non-surgical limb, as well as preferentially load the knee and extensor mechanism relative to the posterior chain muscles [88]. The isometric exercise prescription was implemented as 2 exposures/day, 6–8 h apart, at a load-volume of 3 sets of 45 s per exercise [89]. Starting in weeks 7–8, the front-foot elevated split squat was introduced during the 2nd exposure of the day as an additional 3 sets of knee-dominant isometric exercise; this load volume was maintained (i.e., 3 sets in the 1st exposure, 6 sets in the 2nd exposure) throughout weeks 15–16, with the gradual progression of the double leg squat isometric exercise onto the split squat and wall sit isometric exercises (Table 1). Lastly, the athlete was advised to utilize guidelines from the modified pain-monitoring model and adjust/reduce squat depth if there was any exacerbation of tendon pain (Figure 1) [55,68,72].

#### 2.3.7. Data Collection

Serial data collections were completed throughout the entirety of the athlete’s rehabilitation period [68], with all data collected up to the 16-week timepoint presented in Table 2. Data points were selected based on the standard of care after ACLR and collected as previously described [68]. Primary outcomes of interest were thigh muscle girth (measured 15 cm proximal to the superior pole of the patella) (Figure 3b), lean mass of the lower extremity and quadriceps strength-symmetry/strength relative to bodyweight (Table 2).

## 3. Results

### 3.1. Outcomes

At the 16-week data collection timepoint, the athlete had 2.25 cm more circumferential thigh girth within his surgical (left) limb (relative to his non-surgical limb) (Figure 4) (Figure 5). His InBody 770 multi-frequency bioelectrical impedance analysis (InBody 770, Cerritos, CA, USA) reported an objectively higher lower-extremity lean mass within the surgical limb relative to the non-surgical limb (10.37 kg vs. 10.02 kg, respectively) (Table 2) [90]. Lastly, isometric testing of quadriceps strength on an isokinetic dynamometer (Biodex, Corp., Shirley, NY, USA) yielded a 66% limb symmetry index, with the surgical limb’s average peak torque relative to body weight at 86% (Table 2) [68,91].

### 3.2. Follow-Up

After his 16-week data collection, the athlete continued both clinic- and home-based rehabilitation. He was integrated into a small group strength and conditioning program with ongoing data collection/performance testing as previously published [68]. The athlete completed his return-to-sport training within the same facility as his rehabilitation, and he returned to playing soccer in August of 2022. On 28 September 2022, the athlete returned to his pre-injury level of unrestricted competitive soccer play.

## 4. Discussion

Building upon previous literature, this case report provides an evidence-based framework for the outpatient implementation of BFR after ACLR. Moreover, the integration of BFR into both the clinic- and home-based exercise prescriptions provides a pragmatic rehabilitation strategy for other clinicians to replicate when rehabilitating athletes. The setup, dosing, and progression of exercise for this athlete is justified by the objective outcomes presented in Table 2, and the detailed explanation of the safety precautions utilized for prescribing pBFR during home-based rehabilitation may help to further define an evidence-based approach to pBFR after ACLR.

### 4.1. Safety

Safety is a primary concern when prescribing BFR/pBFR early after ACLR [39,40,92,93]. However, previous literature has reported that the risk of adverse events is extremely low when BFR is prescribed appropriately [39]. To optimize safety and effectiveness, Hughes et al. [39] emphasized the importance of an individualized approach when prescribing partial occlusion pressure for BFR, to which this case report’s BFR/pBFR exercise prescriptions clearly delineate. Wilson et al. [77] reported prescribing pBFR at a 7/10 RPT resulted in partial vascular occlusion in all subjects, suggesting the risk of complete arterial occlusion with this methodology is low. Moreover, the use of patient risk stratification [40], confirmation of partial occlusion with a vascular doppler [39], and the assessment of capillary refill time as a patient self-monitoring strategy exemplifies an evidence-based approach to risk mitigation.

### 4.2. Prescribing Blood Flow Restriction

Previous studies have investigated the effect of high-frequency BFR on TMA after ACLR [28,35,38,47]. Takarada et al. [28] reported significantly less TMA after ACLR when subjects were exposed to BFR (without exercise) 2×/day starting postoperative day 3 through day 14 after ACLR; the observed decrease in quadriceps cross-sectional area was approximately 50% less than their control group. Otah et al. [47] reported that daily low-load exercise with BFR (weeks 2–16) was effective at restoring the cross-sectional area of the quadriceps muscle to the subjects’ preoperative level, and Kilgas et al. [38] investigated the effect of pBFR on quadriceps function within a cohort of subjects with long-standing TMA after ACLR. Collectively, these findings suggest exercise with BFR/pBFR (5–7×/week) is both safe and effective at restoring quadriceps size and strength after ACLR.

Patterson et al. [46] reported that exercise with BFR (2–4 exposures/week) at an intensity of 20–40% of an individual’s 1-RM will likely elicit muscular hypertrophy and strength gains, whereas Arpan Das and Bruce Patons [94] reported that exercise with BFR at 1–20% 1-RM may not exert enough physiological stress to elicit strength adaptations. Considering exercise intensity within the pBFR exercise prescription was constrained to mostly active ROM and bodyweight exercise, a higher exposure to BFR/pBFR was prescribed to maximize the anabolic/anti-catabolic effect of cellular swelling within the surgical limb [28,46,83].

Regarding objective outcomes, this case report provides some novel insights into the anabolic effect of adequate access to twice-daily exercise with BFR/pBFR during the early phase of rehabilitation after ACLR. When implemented for 2 weeks prior to ACLR, Tramer et al. [95] reported no significant effect of high-frequency exercise with BFR (i.e., clinic-based BFR only; 5×/week; 1×/day) on thigh circumference or knee extensor strength. Conversely, the objective data collected at week 16 of this case report suggest that the addition of home-based pBFR (i.e., twice-daily exercise with BFR/pBFR) may facilitate the postoperative recovery of the surgical limb’s thigh girth (beyond the level of the non-surgical limb) (Figure 5). While follow-up time is different between this case report and the aforementioned study (16 weeks vs. 2 weeks, respectively) [95], exercise selection and intensity were similar, and the results suggest an objectively different effect on TMA.

### 4.3. Limb Dominance

Along with the resolution of TMA reported within this case report, it is important to highlight the effect of limb dominance on TMA after ACLR. Strandberg et al. [96] observed the thigh cross-sectional area of the right limb/dominant limb appears to be significantly larger than that of the left/non-dominant limb at baseline. Considering that this athlete’s dominant limb was indeed his right limb, a more pronounced level of TMA may be expected after his left ACLR (i.e., a larger thigh muscle size asymmetry value when using the right/dominant limb’s thigh girth as a reference value); the observation of only 1.5 cm of measurable TMA (compared to the right limb) at 4 weeks after ACLR may be considered a mild amount of TMA within the surgical limb. Moreover, the observation of slightly more lean body mass within the left/surgical limb by postoperative week 16 may be considered an even more impactful outcome when interpreting the effectiveness of this case report’s exercise prescriptions.

### 4.4. Strength Outcome

While this athlete successfully resolved his TMA relatively early after surgery, the home-based pBFR exercise intensity was likely not intense enough to elicit strength gains, and subsequently, strength deficits were still present 16 weeks after ACLR. Adams et al. [97] observed a 25% relative reduction in patellar tendon tensile strength after harvesting the BPTB autograft, which may partially explain the consistent observation of isolated quadriceps weakness after ACLR with the BPTB relative to the hamstring tendon autograft [91,98]. While ongoing reductions in quadriceps strength can impair surgical limb function and delay the achievement of performance testing milestones after ACLR [98,99], recent literature has also observed the restoration of quadriceps strength and between-limb strength symmetry may conversely increase the risk of ACL graft re-injury [99,100,101,102]; this phenomenon may be especially true in athletes who obtain high ACL-Return to Sport after Injury (ACL-RSI) psychometric test scores and when early return to sport progressions are permitted [100,102,103,104]. Along with the advantage of an efficient graft ligamentization process (i.e., graft osteointegration), the presence of isolated quadriceps weakness (a higher hamstrings-to-quadriceps strength ratio within the surgical knee) after ACLR with the BPTB may partially explain the observation of improved graft survivorship relative to the hamstring tendon autograft within cohorts at high risk of ACL reinjury [9,105,106,107,108].

### 4.5. Case Study Limitations

Regarding the resolution of TMA as the result of the exercise interventions found within this case report, there are limitations related to interpreting the effect of the exercise prescriptions. (1) Within other pragmatic interventional studies, low exercise compliance has been reported [109]; using verbal confirmation as a method of compliance tracking may be considered a low-quality methodology relative to more formal strategies (i.e., direct compliance tracking by observation or the use of compliance tracking forms/software). (2) The use of circumferential thigh measurements and bioelectrical impedance analysis to quantify TMA is not as accurate as the use of dual-energy X-ray absorptiometry or reporting thigh muscle cross-sectional area as a primary outcome [110], which are used in more rigorous studies of TMA [17]. (3) The use of multiple exercise modes (exercise with neuromuscular electrostimulation, BFR/pBFR, and yielding isometric exercise) makes it difficult to ascertain the isolated effect of exercise with BFR/pBFR within this case report. (4) Previous studies have recommended prescribing BFR at a percentage of limb occlusion pressure to standardize interventional exposures [46,111], to which this case report did not. (5) The use of the RPT scale is a less specific method of prescribing partial vascular occlusion and presents as a limitation in interpreting the effectiveness of exercise with BFR/pBFR [112].

## 5. Conclusions

Of the various interventions prescribed to mitigate TMA after ACLR, exercise with BFR has become increasingly popular within the rehabilitation setting [35,38,39,40,41,42]. Due to the time and session-frequency constraints of traditional outpatient rehabilitation (i.e., two to three clinic-based sessions/week), achieving an adequate exposure to exercise with BFR is a challenge [28,46,47], especially when the postoperative status requires exercising with an external load of <20% 1-RM [94]. To optimize exposure to exercise with BFR, the introduction of more practical means of dosing BFR/pBFR within the home-based exercise prescription may be warranted; this case report outlines the clinic and home-based exercise prescriptions utilized to objectively resolve TMA within an athlete after ACLR. Moreover, the detailed methodology presented within this case report can be utilized to optimize the safety and effectiveness of pBFR within the home-based exercise prescription.

## Figures and Tables

**Figure 1 healthcare-11-01885-f001:**
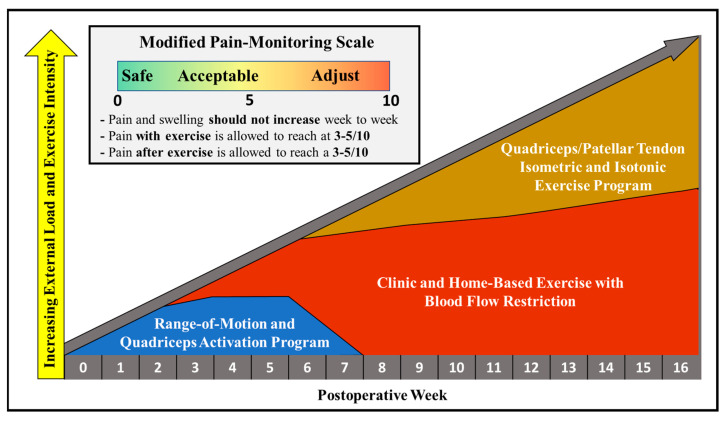
Overview of Clinic- and Home-Based Rehabilitation Program. Description: Over the course of 16 weeks of rehabilitation, the athlete’s clinic and home-based exercise prescriptions were advanced from range of motion exercises during the first 7 weeks of rehabilitation onto load-progressions involving exercise with blood flow restriction and yielding isometrics for the patellar tendon of the surgical limb. A modified pain-monitoring scale was used throughout rehabilitation to guide the advancement of all exercise prescriptions.

**Figure 2 healthcare-11-01885-f002:**
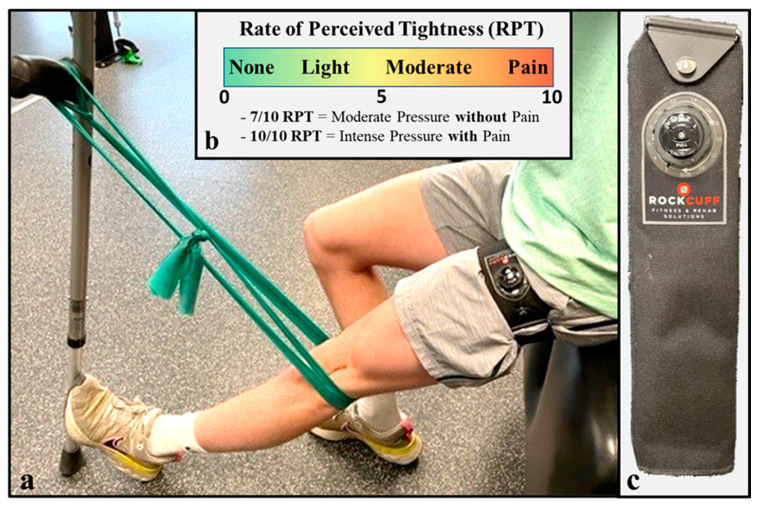
Overview of Practical Blood Flow Restriction Methodology. Description: Exercise with practical blood flow restriction was utilized within the athlete’s home-based exercise prescription to facilitate daily blood flow restriction exposures and optimize the anti-catabolic effect of cellular swelling within the surgical limb. The “crutch quad set” exercise (**a**) was the closed-kinetic-chain base regression within the rehabilitation program. Practical blood flow restriction was prescribed utilizing a rate of perceived tightness (RPT) scale (**b**) and administered with a RockCuff (**c**).

**Figure 3 healthcare-11-01885-f003:**
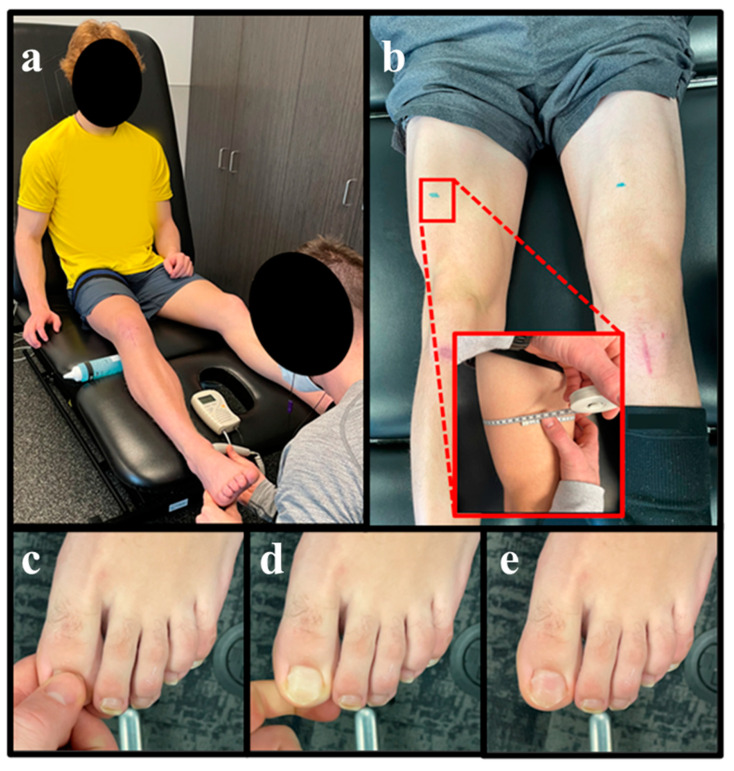
Assessment of Arterial Inflow and Measurement of Circumferential Thigh Girth. Description: During exercise with blood flow restriction, a vascular doppler (**a**) was utilized to confirm the maintenance of resting arterial inflow within the surgical limb. The objective measurement of circumferential thigh girth (**b**) was completed 15 cm proximal to the superior pole of the patella, and a capillary refill time of <3 s (**c**–**e**) was used as an indirect measure of incomplete arterial occlusion (i.e., <100% arterial occlusion).

**Figure 4 healthcare-11-01885-f004:**
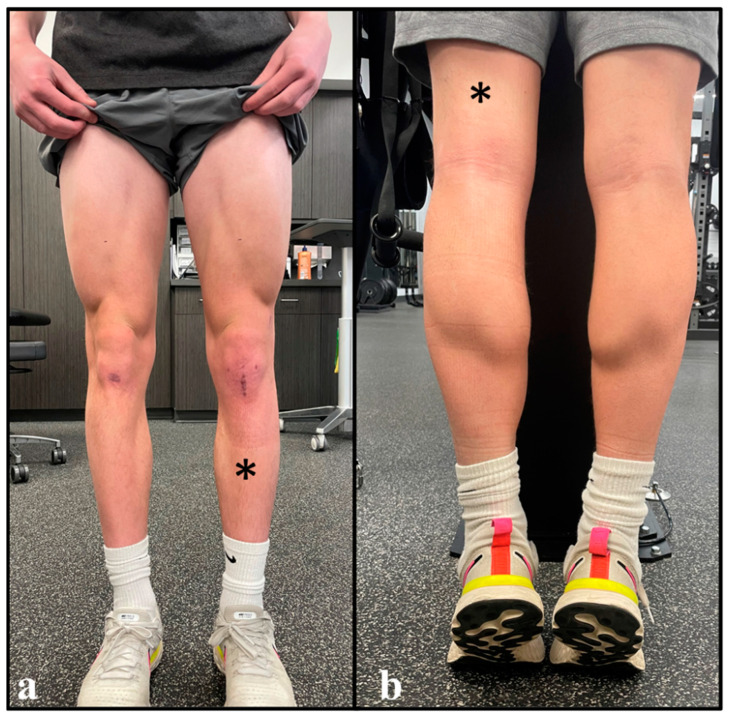
Morphological Appearance of the Lower Extremities 16 Weeks After ACLR. Description: After 16 weeks of clinic- and home-based rehabilitation, objective measurements confirmed the resolution of thigh muscle atrophy within the athlete’s surgical limb (**a**,**b**). Asterisks; surgical limb.

**Figure 5 healthcare-11-01885-f005:**
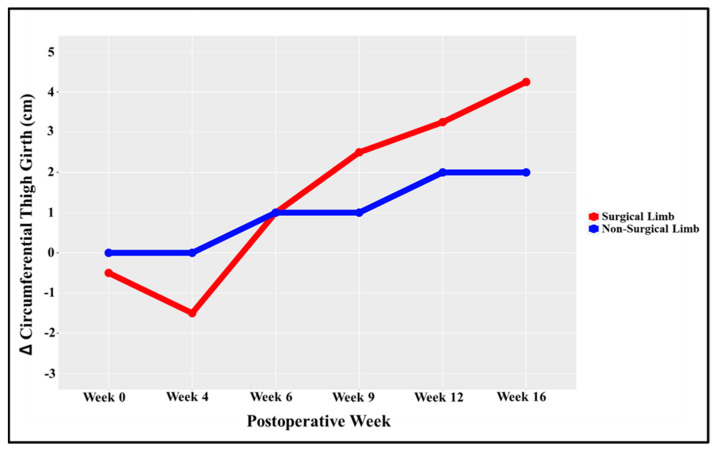
Change in (Δ) Circumferential Thigh Girth. Description: Throughout postoperative weeks 0–16, the athlete efficiently resolved his thigh muscle atrophy and experienced an increase in circumferential thigh girth relative to the non-surgical limb. Y axis; the baseline thigh girth value of the non-surgical limb is used as the “0” point to visualize the between-limb progression of circumferential thigh girth.

**Table 1 healthcare-11-01885-t001:** Clinic and Home-Based Rehabilitation Program (Weeks 0–16).

	Clinic-Based Exercise Prescription (2×/week)	Home-Based Exercise Prescription (Variable Frequency)
**Week 0–1**	- **Quad Set** (2–3 × 10 × 3” Isometric) - **Quad Plank** (2–3 × 45” Isometric) - **Crutch Quad Set** (2–3 × 10 × 3” Isometric) - **Crutch Quad Plank** (2–3 × 45” Isometric) - **Sitting Assisted Heel Slides** (2–3 × 10 × 5” isometric) - **Metronome Quad Setting** (2–3 × 45” at 60–95 BPM) - **NMES** **Quad Set Training** (1 × 15′)	**HEP Frequency**: 5×/day (3–4×/day on clinic-based rehabilitation days)
		- **Quad Set** (2–3 × 10 × 3” Isometric) - **Quad Plank** (2–3 × 45” Isometric) - **Crutch Quad Set** (2–3 × 10 × 3” Isometric) - **Crutch Quad Plank** (2–3 × 45” Isometric) - **Sitting Assisted Heel Slides** (2–3 × 10 × 5” isometric) - **Metronome Quad Setting (2×/day)** (2–3 × 45” at 60–95 BPM) - **Home NMES (2×/day)** **Quad Set Training** (1 × 15′)
**Week 1–2**	- **Quad Set** (2–3 × 8 × 3” Isometric) - **Quad set to SLR** (2–3 × 45” Isometric) - **Crutch Quad Set** (2–3 × 10 × 3” Isometric) - **Stomach Quad Set** (2–3 × 10 × 3” Isometric) - **Stomach Quad Plank** (2–3 × 45” Isometric) - **Metronome Quad setting** (2–3 × 45” at 60–95 BPM) - **BFR** **Crutch Quad Set** (2 × fatigue) **Sitting Assisted Heel Slides** (2 × fatigue)	**HEP Frequency:** 4×/day (3×/day on clinic-based rehabilitation days)
		- **Quad Set** (3 × 10 × 3” Isometric) - **Quad set to SLR** (3 × 45” Isometric) - **Crutch Quad Set** (3 × 10 × 3” Isometric) - **Crutch Quad Plank** (3 × 45” Isometric) - **Sitting Assisted Heel Slides** (3 × 10 × 5” isometric) - **Metronome Quad setting** (3 × 45” at 60–95 BPM) - **Home NMES (2×/day)** **Quad Set Training** (1 × 5–10′) **SLR Training** (1 × 5–10′)
**Week 2–3**	- **Quad set to SLR** (2–3 × 45” Isometric) - **Crutch Quad Set** (2–3 × 10 × 3” Isometric) - **Crutch Quad Plank** (2–3 × 45” Isometric) - **Stomach Quad Set/Hamstring Curl** (2–3 × 10 × 3” Isometric) - **Stomach Quad Plank** (2–3 × 45” Isometric) - **BFR** **Crutch Quad Set** (2 × fatigue) **Sitting Heel Slides** (2 × fatigue) **LAQ** (2 × fatigue)	**HEP Frequency:** 3×/day (2×/day on clinic-based rehabilitation days)
		- **Quad Set to SLR** (2–3 × 45” Isometric) - **Crutch Quad Set** (2–3 × 10 × 3” Isometric) - **Crutch Quad Plank** (2–3 × 45” Isometric) - **Sitting Heel Slides** (2–3 × 30) - **Stomach Quad Set/Hamstring Curl** (2–3 × 10 × 3” Isometric) - **Stomach Quad Plank** (2–3 × 45” Isometric) - **LAQ** (3 × 15 × 3” Isometric) - **Home NMES (2×/day)** **Quad Set Training** (1 × 5′) **SLR Training** (1 × 5′) **LAQ** (1 × 5′)
**Week 3–4**	- **SLR** (2–3 × 45” Isometric) - **Stomach Quad Set/Hamstring Curl** (2–3 × 10 × 3” Isometric) - **Stomach Quad Plank** (2–3 × 45” Isometric) - **Supine Heel Slides** (2 × 45”) - **Hamstring Bridge** (2 × 45”) - **BFR** **Crutch Quad Set** (2 × fatigue) **Sitting Heel Slides** (2 × fatigue) **LAQ** (2 × fatigue)	**HEP Frequency:** 2×/day (1×/day on clinic-based rehabilitation days)
		- **SLR** (2–3 × 45” Isometric) - **Stomach Quad Set/Hamstring Curl** (2–3 × 10 × 3” Isometric) - **Stomach Quad Plank** (2–3 × 45” Isometric) - **Supine Heel Slides** (2 × 45”) - **Hamstring Bridge** (2 × 45”) - **pBFR** **Crutch Quad Set** (2 × fatigue) **Sitting Heel Slides** (2 × fatigue) **LAQ** (2 × fatigue)
**Week 4–5**	- **SLR** (2–3 × 45” Isometric) - **Stomach Quad Set/Hamstring Curl** (2–3 × 10 × 3” Isometric) - **Stomach Quad Plank** (2–3 × 45” Isometric) - **Supine Heel Slides** (2 × 45”) - **Supine Hamstring Bridge** (2 × 45”) - **BFR** **Crutch Quad Set** (2 × fatigue) **Prone Knee Flexion** (2 × fatigue) **LAQ** (2 × fatigue)	**HEP Frequency:** 2×/day (1×/day on clinic-based rehabilitation days)
		- **SLR** (2–3 × 45” Isometric) - **Stomach Quad Set/Hamstring Curl** (2–3 × 10 × 3” Isometric) - **Stomach Quad Plank** (2–3 × 45” Isometric) - **Supine Heel Slides** (2 × 45”) - **Supine Hamstring Bridge** (2 × 45”) - **pBFR** **Crutch Quad Set** (2 × fatigue) **Prone Knee Flexion** (2 × fatigue) **LAQ** (2 × fatigue)
**Week 5–6**	- **SLR** (2–3 × 45” Isometric) - **Supine Heel Slides** (2 × 45”) - **Supine Hamstring Bridge** (2 × 45”) - **BFR** **Crutch Quad Set** (3 × fatigue) **Standing Knee Flexion** (3 × fatigue) **Knee Extension Machine** (3 × fatigue at 30–40 RM)	**HEP Frequency:** 2×/day (1×/day on clinic-based rehabilitation days)
		- **SLR** (2–3 × 45” Isometric) - **Supine Heel Slides** (2 × 45”) - **Supine Hamstring Bridge** (2 × 45”) - **pBFR** **Crutch Quad Set** (2 × fatigue) **Prone Knee Flexion** (2 × fatigue) **LAQ** (2 × fatigue)
**Week 6–7**	- **Standing Banded TKE** (3 × 10 × 3” isometric) - **Banded Squat** (3 × 10) - **Banded Squat Isometric** (3 × 45” isometric at 70 degrees knee flexion) - **FFE Lunge Isometric** (3 × 45” isometric) - **BFR** **Wobble Board Squat** (3 × fatigue) **Standing Knee Flexion** (3 × fatigue) **Knee Extension Machine** (3 × fatigue at 30–40 RM)	**HEP Frequency:** 2×/day (1×/day on clinic-based rehabilitation days)
		- **Standing Banded TKE** (3 × 10 × 3” isometric) - **Banded Squat** (3 × 10) - **Banded Squat Isometric** (3 × 45” isometric at 70 degrees knee flexion) - **pBFR** **Crutch Quad Set** (2 × fatigue) **Prone Knee Flexion** (2 × fatigue) **LAQ** (2 × fatigue)
**Week 7–8**	- **Standing Banded TKE** (3 × 10 × 3” isometric) - **Banded Squat** (3 × 10) - **Banded Squat Isometric** (3 × 45” isometric at 70 degrees knee flexion) - **FFE Lunge Isometric** (3 × 45” isometric) - **BFR** **Wobble Board Squat** (3 × fatigue) **Standing Knee Flexion** (3 × fatigue) **Knee Extension Machine** (3 × fatigue at 20–30 RM)	**HEP Frequency:** 2×/day (1×/day on clinic-based rehabilitation days)
		- **Standing Banded TKE** (3 × 10 × 3” isometric) - **Banded Squat** (3 × 10) - **Banded Squat Isometric** (3 × 45” isometric at 70 degrees knee flexion) - **FFE Lunge Isometric** (3 × 45” isometric) - **pBFR** **Crutch Quad Set** (2 × fatigue) **Prone Knee Flexion** (2 × fatigue) **LAQ** (2 × fatigue)
**Week 8–9**	- **Standing Banded TKE** (3 × 10 × 3” isometric) - **Banded Split-Squat** (3 × 10) - **Banded Split-Squat Isometric** (3 × 45” isometric at 50 degrees knee flexion) - **FFE Lunge Isometric** (3 × 45”) - **BFR** **Wobble Board Squat** (3 × fatigue) **Standing Knee Flexion** (3 × fatigue) **Knee Extension Machine** (3 × fatigue at 20–30 RM)	**HEP Frequency:** 2×/day (1×/day on clinic-based rehabilitation days)
		- **Standing Banded TKE** (3 × 10 × 3” isometric) - **Banded Split-Squat** (3 × 10) - **Banded Split-Squat Isometric** (3 × 45” isometric at 50 degrees knee flexion) - **FFE Lunge Isometric** (3 × 45” isometric) - **pBFR** **Crutch Quad Set** (2 × fatigue) **Prone Knee Flexion** (2 × fatigue) **LAQ** (2 × fatigue)
**Week 9–10**	- **Standing Banded TKE** (3 × 10 × 3” isometric) - **Banded Split-Squat** (3 × 10) - **Banded Split-Squat Isometric** (3 × 45” isometric at 50 degrees knee flexion) - **FFE Lunge Isometric** (2–3 × 45” isometric) - **Wall Sit Isometric** (2–3 × 45” isometric) - **BFR** **Wobble Board Squat** (3 × fatigue) **Single Leg Press** (3 × fatigue at 20–30 RM) **Knee Extension Machine** (3 × fatigue at 20–30 RM)	**HEP Frequency:** 2×/day (1×/day on clinic-based rehabilitation days)
		- **Standing Banded TKE** (3 × 10 × 3” isometric) - **Banded Split-Squat** (3 × 10) - **Banded Split-Squat Isometric** (3 × 45” at 50 degrees knee flexion) - **FFE Lunge Isometric** (3 × 45”) - **pBFR** **Crutch Quad Set** (2 × fatigue) **Prone Knee Flexion** (2 × fatigue) **LAQ** (2 × fatigue)
**Week 10–11**	- **Standing Banded TKE** (3 × 10 × 3” isometric) - **Banded Split-Squat** (2–3 × 10) - **Banded Split-Squat Isometric** (2–3 × 45” isometric at 50 degrees knee flexion) - **Wall Sit Isometric** (2–3 × 45” isometric) - **BFR** **Single Leg Press** (3 × fatigue at 20–30 RM) **Standing Knee Flexion Machine** (3 × fatigue at 15–20 RM) **Knee Extension Machine** (3 × fatigue at 20–30 RM)	**HEP Frequency:** 2×/day (1×/day on clinic-based rehabilitation days)
		- **Standing Banded TKE** (3 × 10 × 3” isometric) - **Banded Split-Squat** (2–3 × 10) - **Banded Split-Squat Isometric** (2–3 × 45” isometric at 50 degrees knee flexion) - **FFE Lunge Isometric** (2–3 × 45” isometric) - Wall Sit Isometric (2–3 × 45” isometric) - **pBFR** **Crutch Quad Set** (2 × fatigue) **Prone Knee Flexion** (2 × fatigue) **LAQ** (2 × fatigue)
**Week 11–12**	- **Banded Split-Squat** (2–3 × 10) - **Banded Split-Squat Isometric** (2–3 × 45” isometric at 50 degrees knee flexion) - **Wall Sit Isometric** (2–3 × 45” isometric) - **Suitcase Lunges** (2–3 × 10 yards × 25# DB each way) - **BFR** **Single Leg Press** (3 × fatigue at 20–30 RM) **Standing Knee Flexion Machine** (3 × fatigue at 15–20 RM) **Knee Extension Machine** (3 × fatigue at 20–30 RM)	**HEP Frequency:** 2×/day (1×/day on clinic-based rehabilitation days)
		- **FFE Lunge Isometric** (2–3 × 45” isometric) - **Wall Sit Isometric** (2–3 × 45” isometric) - **pBFR** **Split Squat** (2 × fatigue) **Standing Knee Flexion** (2 × fatigue) **LAQ** (2 × fatigue)
**Week 12–13**	- **Banded Split-Squat** (2–3 × 10) - **Banded Split-Squat Isometric** (2–3 × 45” isometric at 50 degrees knee flexion) - **Wall Sit Isometric** (2–3 × 45” isometric) - **Suitcase Lunges** (2–3 × 10 yards × 25# DB each way) - **BFR** **Single Leg Press** (3 × fatigue at 20–30 RM) **Standing Knee Flexion Machine** (3 × fatigue at 15–20 RM) **Knee Extension Machine** (3 × fatigue at 20–30 RM)	**HEP Frequency:** 2×/day (1×/day on clinic-based rehabilitation days)
		- **FFE Lunge Isometric** (2–3 × 45” isometric) - **Wall Sit Isometric** (2–3 × 45” isometric) - **pBFR** **Split Squat** (2 × fatigue) **Standing Knee Flexion** (2 × fatigue) **LAQ** (2 × fatigue)
**Week 13–14**	- **Banded Split-Squat** (2–3 × 10) - **Banded Split-Squat Isometric** (2–3 × 45” isometric at 70 degrees knee flexion) - **Wall Sit Isometric** (2–3 × 45” isometric) - **Suitcase Lunges** (2–3 × 10 yards × 25# DB each way) - **BFR** **Single Leg Press** (3 × fatigue at 15–20 RM) **Standing Knee Flexion Machine** (3 × fatigue at 10–15 RM) **Knee Extension Machine** (3 × fatigue at 15–30 RM)	**HEP Frequency:** 2×/day (1×/day on clinic-based rehabilitation days)
		- **FFE Lunge Isometric** (2–3 × 45” isometric) - **Wall Sit Isometric** (2–3 × 45” isometric) - **pBFR** **Split Squat** (2 × fatigue) **Standing Knee Flexion** (2 × fatigue) **LAQ** (2 × fatigue)
**Week 14–15**	- **Banded Split-Squat** (2–3 × 10) - **Banded Split-Squat Isometric** (2–3 × 45” at 70 degrees knee flexion) - **Wall Sit Isometric** (2–3 × 45”) - **Suitcase Lunges** (2–3 × 10 yards × 35# DB each way) - **BFR** **Single Leg Press** (3 × fatigue at 15–20 RM) **Standing Knee Flexion Machine** (3 × fatigue at 10–15 RM) **Knee Extension Machine** (3 × fatigue at 15–30 RM)	**HEP Frequency:** 2×/day (1×/day on clinic-based rehabilitation days)
		- **FFE Lunge Isometric** (2–3 × 45” isometric) - **Wall Sit Isometric** (2–3 × 45” isometric) - **pBFR** **Split Squat** (2 × fatigue) **Standing Knee Flexion** (2 × fatigue) **LAQ** (2 × fatigue)
**Week 15–16**	- **Banded Split-Squat** (2–3 × 10) - **Banded Split-Squat Isometric** (2–3 × 45” isometric at 70 degrees knee flexion) - **Wall Sit Isometric** (2–3 × 45” isometric) - **Suitcase Lunges** (2–3 × 10 yards × 35# DB each way) - **BFR** **Single Leg Press** (3 × fatigue at 15–20 RM) **Standing Knee Flexion Machine** (3 × fatigue at 10–15RM) **Knee Extension Machine** (3 × fatigue at 15–30 RM)	**HEP Frequency:** 2×/day (1×/day on clinic-based rehabilitation days)
		- **FFE Lunge Isometric** (2–3 × 45” isometric) - **Wall Sit Isometric** (2–3 × 45” isometric) - **pBFR** **Split Squat** (2 × fatigue) **Standing Knee Flexion** (2 × fatigue) **LAQ** (2 × fatigue)

Prescription parameters, (sets(×)reps(×)isometric hold time); quad; quadriceps; set, 3 s isometric hold; plank, 45 s isometric hold; single quotation mark, minutes; double quotation mark, seconds; fatigue, performance of sets to volitional muscle fatigue; metronome quad setting; use of a metronome to cue on/off quadriceps contractions; NMES, neuromuscular electrostimulation; HEP, home exercise program; SLR, straight-leg raise; BFR, exercise with blood flow restriction; LAQ, long-arc quadriceps/sitting knee extensions; hamstring bridge; long-lever supine hamstring/glute bridge with knees in extension; pBFR, practical blood flow restriction; TKE, terminal knee extension; FFE, front foot elevated; RM, repetition maximum; suitcase lunges; weighted forward lunge with a single dumbbell in one hand; #, pounds.

**Table 2 healthcare-11-01885-t002:** Objective Data and Outcomes.

Objective Measurement		Week 0	Week 4	Week 6	Week 9	Week 12	Week 16
Effusion (0 –3+ Sweep Test)	Surgical	3+	3+	2+	1+	1+	Trace
Thigh Girth (cm)	Surgical	47.5	46.5	49	50.5	51.25	52.25
	Non-Surgical	48	48	49	49	50	50
	Difference	−0.5	−1.5	0	1.5	1.25	2.25
Calf Girth (cm)	Surgical	35	35	35	36	36.25	36.5
	Non-Surgical	35.25	35.25	35.25	35.5	35.5	35.5
	Difference	−0.25	−0.25	−0.25	0.75	0.75	1
Range of Motion (degrees)	Surgical	5-0-30	5-0-125	5-0-130	5-0-130	8-0-145	8-0-150
	Non-Surgical	8-0-150	8-0-150	8-0-150	8-0-150	8-0-150	8-0-150
Straight-Leg Raise (degree lag)	Surgical	5	0	0	0	0	0
Quadriceps Strength/BW (%)	Surgical	NA	NA	NA	NA	70	87
	Non-Surgical	NA	NA	NA	NA	96	135
	LSI	NA	NA	NA	NA	73	66
KT-1000 (mm)	Surgical	NA	NA	NA	5,6,7	4,5,6	4,5,6
	Non-Surgical	NA	NA	NA	5,6,7	5,6,7	5,6,7
Lean Mass Analysis (kg)	Surgical	NA	NA	NA	NA	NA	10.37
	Non-Surgical	NA	NA	NA	NA	NA	10.02
	Difference	NA	NA	NA	NA	NA	0.35

Week; data collected at the start of the postoperative week; mm, millimeters; cm, centimeters; kg, kilograms; thigh girth, circumferential thigh girth measured 15 cm proximal to the superior pole of the patella; calf girth, circumferential calf girth recorded as the largest value measured within the proximal half of the lower limb; ROM; knee range of motion measured with the goniometer; SLR, straight-leg raise test recorded as the degree of knee lag into flexion as the limb is lifted off the supporting surface; quadriceps strength/BW, measurement of knee extension strength in kilograms on an isokinetic dynamometer relative to body weight in kilograms; KT, measurements of ACL graft laxity with a KT-1000 arthrometer; lean mass analysis, lean mass in kilograms as collected on an InBody 770 multi-frequency bioelectrical impedance device.

## Data Availability

All data related to this case report are available upon request. Please contact the corresponding author.

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
