# Peer review of "Clinic and Home-Based Exercise with Blood Flow Restriction Resolves Thigh Muscle Atrophy after Anterior Cruciate Ligament Reconstruction with the Bone-Patellar Tendon-Bone Autograft: A Case Report"

_healthcare, 2023, doi:10.3390/healthcare11131885_

Round 1

Reviewer 1 Report

This study reports that exercise with blood flow restriction during clinic and home-based rehabilitation could resolve thigh muscle atrophy after anterior cruciate ligament reconstruction. The present work is of interest and some aspects are perhaps already known, but it's valuable work. The study design is novel. Follow up duration is proper. It's characterized by a good order, clear and very didactic approach.     It's a comprehensive study. The conclusions are consistent with the evidence and arguments and address the main question. The references are appropriate.

I suggest the authors make some revisions of content and structure. I propose change the title, it's too long. 

Minor editing of English language required. 

Author Response

Comment 1

This study reports that exercise with blood flow restriction during clinic and home-based rehabilitation could resolve thigh muscle atrophy after anterior cruciate ligament reconstruction. The present work is of interest and some aspects are perhaps already known, but it's valuable work. The study design is novel. Follow up duration is proper. It's characterized by a good order, clear and very didactic approach. It's a comprehensive study. The conclusions are consistent with the evidence and arguments and address the main question. The references are appropriate.

I suggest the authors make some revisions of content and structure. I propose change the title, it's too long.

Author’s response:

Thank you for your suggestions, as we (authors) agree with your concern over the length of the title. The title now reads, “Clinic and Home-Based Exercise with Blood Flow Restriction Resolves Thigh Muscle Atrophy after Anterior Cruciate Ligament Reconstruction with the Bone-Patellar Tendon-Bone Autograft: A Case Report.”

Reviewer 2 Report

The case study is well presented and described, it can be an added value to the journal. I only suggest summarizing the methods section, many information in this section could be omitted. The table 1 could be added in the supplemental data.

Reviewer 3 Report

2 The role of High-Frequency Exercise with Blood Flow Restriction in Clinic and Home-Based Rehabilitation on Anterior Cruciate Ligament Reconstruction-related Thigh Muscle Atrophy: A Case Report

13-17 I would reformulate this part because if it is already widespread for the ACL-R the drafting of your manuscript (case report) does not seem necessary.. more than anything else it seems a novelty to evaluate the impact of the home BFR.

21 I recommend reporting data, with instrumental scales that unfortunately you don't describe

29-30 Anterior cruciate ligament (ACL) rupture is commonly related to dynamic knee valgus (DKV) as an erratic motion pattern, recognized as a risk factor for joint stress. (REF: Marotta N, Demeco A, de Scorpio G, Indino A, Iona T, Ammendolia A. Late Activation of the Vastus Medialis in Determining the Risk of Anterior Cruciate Ligament Injury in Soccer Players. J Sport Rehabil. 2019 Nov 7;29(7):952-955. doi: 10.1123/jsr.2019-0026. PMID: 31711040.)

76 "the result of direct contact to the lateral trunk during competitive play" the mechanism is not clear

84injured limb was completed and confirmed "normal", .. normal is not a clinical term, it indicates the absence or presence of disturbances.

87 McMurray!

159 is the figure yours?

208 I would focus a lot on the concept of home-based, on its security, being quite original as an approach.

Minor editing of English language required

Round 2

Reviewer 3 Report

Dear authors, I can consider the adequacy of your manuscript, but I must necessarily point out to the editor that a case report cannot provide such clear-cut conclusions and recommendations:

"L508 Of the various interventions prescribed to mitigate TMA after ACLR, exercise with BFR should be considered a primary intervention."

You can suggest safety and adequacy, but not that it should be considered as a primary intervention.
